# OpenReview forum: "LLM Agents Should Employ Security Principles"
_NeurIPS.cc/2025/Position_Paper_Track — Submitted to NeurIPS 2025 Position Paper Track_

### Official Review · Reviewer_MBFM · 2025-07-22

**Significance:** 3
**Presentation:** 3
**Rating:** 9
**Confidence:** 4

**Summary:**

This position paper highlights that current LLM agents face serious, unresolved security and privacy risks, especially in multi-agent scenarios and under prompt injection attacks. The authors argue for explicitly adopting established security principles—defense-in-depth, least privilege, complete mediation, and psychological acceptability—rather than relying on ad hoc defenses. They propose AgentSandbox, a framework that systematically incorporates these principles through component isolation, context-aware access control, and adaptive policy management. Empirical results on the AgentDojo benchmark show that AgentSandbox offers a better balance of security and utility than existing methods. The paper calls on the NeurIPS community to make such foundational security principles standard practice for LLM agents.

**Strengths:**

1. Clear, actionable agenda: Provides a practical security agenda for agentic AI, bridging decades of security engineering wisdom and modern LLM challenges.

2. Novel system architecture: The AgentSandbox design (with persistent/ephemeral agents, I/O firewalls, etc.) is original, modular, and extensible.

3. Empirical grounding: Evaluations are performed on AgentDojo with multiple LLMs, using multiple baselines, supporting the position with real data.

**Weaknesses:**

1. Conceptual/prototype level: While AgentSandbox is well-motivated, it remains a conceptual framework; some implementation details are high-level or left for future work.

2. Evaluation scope: Empirical evaluation, though strong, is mostly on synthetic benchmarks (AgentDojo); further deployment in real-world, long-running agent settings would strengthen claims.

3. Adaptability and overhead: Some discussion of computational/resource overhead or usability trade-offs in complex multi-agent environments would be valuable.

**Questions:**

1. How does AgentSandbox perform when scaled to real-world, persistent multi-agent systems with dynamic, evolving tasks and long-term memory?

2. What are the expected computational and operational overheads of maintaining ephemeral agents and continuous mediation in production?

3. Can the reward modeling policy engine adapt to new, unseen attack vectors without explicit rule updates, or is continual retraining required?

**Alternative Position:**

Yes, and alternative positions are well-considered and addressed by the argument

**Author Identification:**

No.

**Context:**

3

**Discussion:**

4

**Ethics:**

["NO or VERY MINOR ethics concerns only"]

**Position:**

Yes, the paper argues for or against a position related to machine learning.

**Support:**

4

**Thoroughness:**

4

---

### Official Review · Reviewer_QPYc · 2025-08-06

**Significance:** 4
**Presentation:** 3
**Rating:** 7
**Confidence:** 4

**Summary:**

This paper propose a position: the well-established design principles in information security should be employed when deploying LLM agents at scale. To justify the position, the authors propose a security framework called AgentSandbox that applies these security principles into the agents. They further provide illustrative examples and evaluations to show the resistence of AgentSandbox againts privacy and security problems.

**Strengths:**

The security of LLM agents is a very important topic and this work is very timely.
The proposed position is very reasonable, especially the idea of incorporating well-established secutiry principles from studies of  information security.
The proposed framework is well described and shows a lot of potential. While this work only shows a basic impelmentation of these principles, the framework indeed brings many inspirations for future development of more robust and trustworthy agents.

**Weaknesses:**

I only have a minor concern. Since this work already proposes a security framework, what other aspects should future agent builder and deployer be care of?

**Questions:**

While I am not an expert in information security, I am curious that except for the 4 principles discussed by this work, what about other principles, since the authors mention that '... introduced eight design principles for secure systems...'?

The proposd framework is described in a general way, then are there any specific challenges when applying it to different domains, like medical, finance etc? I am curious in more details.

**Alternative Position:**

No

**Author Identification:**

No.

**Context:**

3

**Discussion:**

4

**Ethics:**

["NO or VERY MINOR ethics concerns only"]

**Position:**

Yes, the paper argues for or against a position related to machine learning.

**Support:**

4

**Thoroughness:**

5

---

### Official Review · Reviewer_gKAX · 2025-08-26

**Significance:** 4
**Presentation:** 4
**Rating:** 8
**Confidence:** 3

**Summary:**

"LLM Agents Should Employ Security Principles" argues that LLM agent applications should adopt a general framework that ensures cybersecurity by design, drawing inspiration from traditional security principles. The authors propose a specific design pattern representing their own interpretation of how those traditional principles should be translated into the novel domain of LLM agents. While the overall goal of cybersecure agents is not a controversial position, there is not yet a level of consensus on the appropriate design patterns as there is in more established domains of cybersecurity. Discussing what a good design pattern would look like is an excellent and timely topic.

The authors advocate for a separation of permanent and ephemeral AI agents to compartmentalize the application, thereby creating barriers around potential penetration vectors. They also envision a firewall for external traffic and an additional layer of protection between the user data and the ephemeral agent. In doing so, they incorporate traditional principles such as least-privilege data access, complete mediation on every agent action, defense-in-depth, while also following psychological acceptability by minimizing user intercession.

**Strengths:**

While anyone would typically agree that LLM agent applications should be cybersecure, it is not clear exactly what design patterns should become the industry-standards the way developers have such standards for other domains, such as internet protocols, and OAuth. Determining what these design patterns would be is a major undertaking, but a timely one as agentic applications come to market and new cybersecurity threats emerge. Security should be a concern to all developers and researchers of LLMs, and a venue like NeurPS is an excellent one to discuss these principles.
The article does not simply state that agentic applications should be secure, but goes further by introducing their own proposed design pattern complete with demonstration on a benchmark. Demonstrations with measurable outcomes like this are a very fruitful way to commence a discussion: no one work will arrive at a completely full-proof principle, but attempts like this are sure to stimulate the kinds of discussions that can lead us there.

**Weaknesses:**

The core position that agentic LLMs should use security principles is hard to disagree with as a high-level idea and so I think the more fruitful area of discussion is what colleagues think of the particular design pattern proposed with a view to constructively addressing weaknesses and agreeing on strengths.

**Questions:**

This paper provides a great applied example with the AgentDojo benchmark. This is sufficient for the paper's goal of a proof of concept, however surely more comprehensive testing is warranted before trusting such an application in the real world. My question is: do the authors have any insight into how, in the future, developers could comprehensively test a design like that of AgentSandbox against enough possible threats as to make it production ready?

**Alternative Position:**

Yes, and alternative positions are well-considered and named but not addressed

**Author Identification:**

No.

**Context:**

4

**Discussion:**

4

**Ethics:**

["NO or VERY MINOR ethics concerns only"]

**Position:**

Yes, the paper argues for or against a position related to machine learning.

**Support:**

4

**Thoroughness:**

4

---

### Note · Authors · 2025-09-04

**1-11 Submit Again:**

Definitely yes

**1-1 Submission Process:**

5

**1-2 Next Year:**

We would encourage a stronger emphasis on papers that not only identify challenges but also propose and debate concrete, actionable frameworks. While identifying problems is critical, the community would greatly benefit from forward-looking position papers that explore tangible pathways for translating our principles into practice.

**1-3 Future Development:**

We would suggest other than standard presentations, group accepted papers by topic and host moderated panel discussions among the authors.

**1-4 Interest:**

["Panel discussions with other position paper authors", "Structured debates on controversial topics", "Workshops for developing position papers", "Mentorship programs for early-career researchers"]

**1-5 Thoughtful:**

10

**1-6 Supportive:**

10

**1-7 Technical Aspects Versus Position:**

8

**1-8 Gate Keeping:**

9

**1-9 Camera Ready Changes:**

We will expand our discussion on designing and deploying secure agents in production environments. The primary goal of this position paper is to advocate for the adoption of a security mindset in the design and evaluation of LLM agents, as many well-established security principles are not yet systematically applied in this domain. By clarifying these points, we aim to better equip the community to shape the future of secure agentic systems.

**3-1 Review Response1:**

gKAX

**3-2 Reaction To Review1:**

We are happy that the reviewer found our work a timely and major undertaking, and they believe our proposed design pattern with its benchmark demonstration is a fruitful way to stimulate the discussion toward industry-standard solutions, and for highlighting that NeurIPS is an excellent venue to discuss these important principles.

We found this review to be exceptionally thoughtful and supportive, as it focused on the technical merits of our work and provided constructive, forward-looking questions.

We address their concerns below:

**Further discussion on deploying AgentSandbox in production**:
We would like to highlight that the point of our paper's primary goal is to advocate for the adoption of a security mindset in the design and evaluation of LLM agents, as many well-established security principles are not yet common knowledge in this domain. A production-ready system cannot be achieved through a fixed set of tests, but rather through a continuous process of design and evaluation guided by this mindset. Having these principles in mind will help shape future designs.

In the past, a sandbox [1] test-bed, as defined by NIST, was designed to prevent potentially malicious software from accessing any system resources beyond those for which it is explicitly authorized. Our approach is deeply rooted in the history of computer security. Therefore, comprehensively testing a design like ours involves validating the robustness of these isolation mechanisms and ensuring that the agent's permissions are restricted to the essential set required for a given task, in alignment with foundational security principles. We also included other discussions in Appendix E.

Reference
[1]. Sandbox, https://csrc.nist.gov/glossary/term/sandbox, NIST.

**3-3 Review Response2:**

QPYc

**3-4 Reaction To Review2:**

We are happy that the reviewer found our work on this timely and important topic to be very reasonable, and that they believe our proposed framework is well-described, shows a lot of potential, and provides inspiration for future development.

We appreciate this constructive and highly supportive review, which correctly identifies the potential of our work and raises insightful questions regarding its broader applicability and future directions.

We address their concerns below:

**Discussion on future deployer considerations**: Future agent builders should adopt a security mindset rather than relying on a fixed checklist of features. While our framework instantiates this mindset, its underlying principles are orthogonal to any single architecture. Our paper focuses on personal assistant agents, but the security considerations in other domains can be fundamentally different.

**Further details of other principles**: We focused on four principles identified as the minimum necessary foundation for the AgentSandbox setting. We agree that other classical principles are equally important, though their primary applications differ. For instance, open-design is widely used in domains like cryptography; fail-safe defaults are critical for monitoring and control systems; separation of privilege prevents a single point of compromise; and minimizing the attack surface is a vital, complementary principle that informs the initial design phase.

**Discussion on other domains**: Applying our framework to high-stakes domains introduces unique challenges tied to specific risk profiles and regulatory landscapes. In medicine, the focus on patient safety and privacy necessitates integrating security controls with clinical reasoning to ensure agent actions are both authorized and medically sound. In finance, the framework must be adapted to safeguard financial integrity and meet stringent auditability requirements, necessitating immutable logging of all agent actions for compliance.

**3-5 Review Response3:**

MBFM

**3-6 Reaction To Review3:**

We are glad the reviewer found our work to have a clear and actionable agenda, a novel system architecture, and strong empirical grounding.

We appreciate the reviewer for this insightful and highly constructive feedback, which accurately identifies our core technical contributions while raising important, forward-looking questions about the system's scalability and operational trade-offs.

We address their concerns below:

**Clarification on AgentSandbox scalability**: We clarify that our evaluation is on the AgentDojo benchmark, which provides a standardized framework of 97 realistic tasks and 629 security test cases across four suites: Workspace (emails, calendar, cloud drive, etc), Slack (messaging, web browsing, file access, etc), Banking (transactions, statements, etc), and Travel (flights, restaurants, car rentals, etc). Future works can focus on developing efficient mechanisms for dynamic privilege management to handle evolving tasks and extending to long-term memory and inter-agent communication to prevent data leakage and collusion.

**Computational overhead**: The overhead is minimal. We would like to clarify that each instantiated ephemeral agent for the task which is terminated on completion, eliminating state maintenance costs. In modern agentic frameworks like LangChain and AutoGen which are designed for rapid agent creation and disposal. Consequently, the computational overhead is not a fixed cost of the framework itself but is determined entirely by the task's complexity and its specific requests.

**Discussion on adapting reward modeling policy engine**: The policy engine generalizes to unseen attacks within a trained domain (e.g., flights, car rentals in “Travel”) but requires retraining for fundamentally different domains (e.g., “Slack” v.s. “Banking”). While continual retraining for every new vector is not required, deploying in a new operational domain necessitates a new training process to learn the context, security requirements, etc.

---

### Meta-Review · Area_Chair_M1Mf · 2025-09-02

**Rating:** 8
**Confidence:** 4

**Strengths:**

The paper makes a strong, timely, and impactful contribution by advocating for security-by-design in LLM agents and proposing a concrete framework (AgentSandbox) to illustrate how such principles can be operationalized. While the evaluation is limited to benchmarks and some implementation details remain conceptual, the contribution is highly relevant to the NeurIPS community and sets the stage for further research.

**Weaknesses:**

For improvement, the authors are encouraged to discuss additional security principles and how they apply to LLM agents, and provide more detail on computational overhead and scalability in practical deployments.

**Questions:**

no questions

**Ethics:**

no ethics issues

**Thoroughness:**

2

---

### Decision · Program_Chairs · 2025-09-26

Reject